

# The evolution of reproductive strategies in turtles

Gabriel Jorgewich-Cohen[1], Rafael S. Henrique[2], Pedro Henrique Dias[3] and Marcelo R. Sánchez-Villagra[1]

[1] Paläontologisches Institut und Museum, Universität Zürich, Zürich, Zürich, Switzerland
[2] Laboratório de Anfíbios, Instituto de Biociências, Universidade de São Paulo, São Paulo, São Paulo, Brazil
[3] Departamento de Zoologia, Universidade Federal do Paraná, Curitiba, Paraná, Brazil

## ABSTRACT

Optimal egg size theory assumes that changes in the egg and clutch are driven by selection, resulting in adjustments for the largest possible production of offspring with the highest fitness. Evidence supports the idea that large-bodied turtles tend to produce larger clutches with small and round eggs, while smaller species produce small clutches with large and elongated eggs. Our goals were to investigate whether egg and clutch size follow the predictions of egg size theory, if there are convergent reproductive strategies, and identify ecological factors that influence clutch and egg traits across all clades of living turtles. Using phylogenetic methods, we tested the covariance among reproductive traits, if they are convergent among different turtle lineages, and which ecological factors influence these traits. We found that both egg shape and size inversely correlate with clutch size, although with different evolutionary rates, following the predictions of the egg size theory. We also present compelling evidence for convergence among different turtle clades, over at least two reproductive strategies. Furthermore, climatic zone is the only ecological predictor to influence both egg size and fecundity, while diet only influences egg size. We conclude that egg and clutch traits in Testudines evolved independently several times across non-directly related clades that converged to similar reproductive strategies. Egg and clutch characteristics follow the trade-offs predicted by egg size theory and are influenced by ecological factors. Climatic zone and diet play an important role in the distribution of reproductive characteristics among turtles.

## INTRODUCTION

Macroevolutionary patterns in amniote reproduction (*Battistella et al., 2019*; *Murray, Crother & Doody, 2020*; *Starck, Stewart & Blackburn, 2021*) can be investigated based on the diversity of traits in egg and clutch (*e.g.*, *Kaplan & Salthe, 1979*; *Deeming & Birchard, 2007*; *Jetz, Sekercioglu & Böhning-Gaese, 2008*; *Deeming & Ruta, 2014*). The idea of an "optimal" correlation between egg and clutch size, based on trade-offs associated to K/r strategies, has led to several discussions without a consensus about the distribution or reasons of such correlations (*Smith & Fretwell, 1974*; *Congdon & Gibbons, 1987*; *Wilbur & Morin, 1988*; *Elgar & Heaphy, 1989*; *Godfray, Partridge & Harvey, 1991*; *Kuchling, 1999*;

Corresponding author
Gabriel Jorgewich-Cohen,
gabriel.jorgewichcohen@pim.uzh.ch

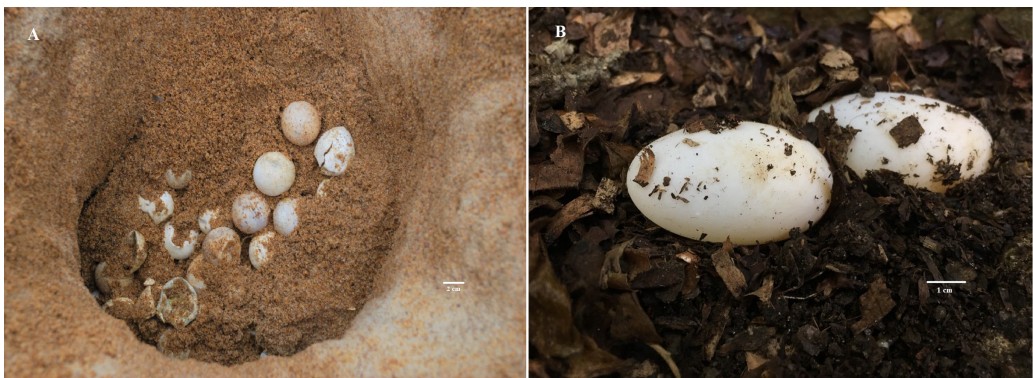

**Figure 1 Egg and clutch strategies.** Examples of different strategies: nest of the giant Arrau turtle (*Podocnemis expansa*) with many small round eggs (A); small clutch with big and elongated eggs of the South American wood turtle (*Rhinoclemmys punctularia*) (B). The adult carapace length of these two species reaches over 1 m and 25 cm long, respectively.

*Zhao, Chen & Liao, 2017*; *Yu & Deng, 2020*). Optimal egg/clutch size theory assumes that changes in the egg and clutch are driven by selection, resulting in adjustments for the largest possible production of offspring with the highest fitness, at the lowest cost to their progenitors (*Brockelman, 1975*; *Congdon & Gibbons, 1987*; *Janzen & Warner, 2009*).

Turtles offer a rich subject of investigation, given the ecological diversity of the group. Studies focused on turtles have tested many correlations between egg size and both morphological and ecological traits in an effort to explain the variation among species (*Elgar & Heaphy, 1989*; *Iverson, 1992*; *Iverson et al., 1993*; *Iverson, Lindeman & Lovich, 2019*; *Rowe, 1994*; *Rachmansah, Norris & Gibbs, 2020*). Some authors have argued that the "optimum" egg size is determined by adult body size (*Gibbons, 1982*), pelvic aperture morphology (*Congdon & Gibbons, 1987*; *Kuchling, 1999*; *Clark, Ewert & Nelson, 2001*; *Hofmeyr, Henen & Loehr, 2005*), environmental factors, such as resource availability and temperature influenced by habitat and biogeography (*Hofmeyr, Henen & Loehr, 2005*; *Macip-Ríos et al., 2012*; *Macip-Ríos, Sustaita-Rodriguez & Casas-Andreu, 2013*), phylogenetic distribution and/or physiology (*Bowden et al., 2004*, *Cordero, 2021*).

Evidence supports the idea that large-bodied turtles tend to produce larger clutches with relatively small and round eggs (Fig. 1A), while smaller species produce small clutches with relatively large and elongated eggs (Fig. 1B). *Elgar & Heaphy (1989)* proposed that spherical eggs are less susceptible to desiccation as the surface-volume ratio is smaller in comparison to elongated eggs—therefore being more suitable for warmer areas. In contrast, *Pritchard (1979)* suggested that small species tend to produce bigger, elongated eggs because a small spherical egg would not be capable of producing a functional hatchling due to a lack of space, and that adult body size is a constraint for egg width. *Moll (1979)* argued that spherical eggs occupy space more efficiently than elongated eggs, thereby allowing the fit of larger clutches in the abdominal cavity.

Many trends in egg and clutch characteristics also seem to be influenced by ecological factors. *Rachmansah, Norris & Gibbs (2020)* suggested that the broad access to resources in tropical areas, supports larger-bodied taxa to produce more eggs. *Craven et al. (2008)*

**Table 1 Hierarchical models of evolutionary correlation among reproductive traits in turtles.**

| Model | $\sigma^2$ 1,1 | $\sigma^2$ 1,2 | $\sigma^2$ 1,3 | $\sigma^2$ 2,1 | $\sigma^2$ 2,2 | $\sigma^2$ 2,3 | R1 | R2 | R3 | Log(L) | AIC |
|---|---|---|---|---|---|---|---|---|---|---|---|
| Common rates, common correlation | 0.002 | – | – | 0.0001 | – | – | 0.519 | – | – | 199.12 | −388.25 |
| Different rates, common correlation | **0.0043** | **0.0015** | **0.0007** | **0.0002** | **0.0001** | **0** | **0.567** | **–** | **–** | **222.99** | **−427.99** |
| Common rates, different correlation | 0.0021 | – | – | 0.0001 | – | – | 0.356 | 0.599 | 0.958 | 209.02 | −404.05 |
| No common structure | 0.004 | 0.0015 | 0.001 | 0.0002 | 0.0001 | 0 | 0.458 | 0.586 | 0.829 | 224.11 | −426.22 |

Note:
Model description, rates of correlation between egg size and three different clutch size groups ($\sigma^2$ 1,x), rates of correlation between egg shape and three different clutch size groups ($\sigma^2$ 2,x), correlation between egg size and egg shape, affected by different regimes of clutch size (R), log-likelihood (Log-L), and Akaike information criterion (AIC) for four multivariate Brownian evolution model fits to egg and clutch data. The best-supported model is highlighted in bold.

proposed that resource availability and type of diet might play a role in egg nutrition (*Craven et al., 2008*). *Spencer & Janzen (2011)*, advocate that higher mean temperature of tropical areas may influence embryo metabolism and favor earlier hatching—favoring the production of more clutches per year.

Although general trends have been identified (*e.g.*, *Iverson, Lindeman & Lovich, 2019*; *Rachmansah, Norris & Gibbs, 2020*), a comprehensive analysis exploring egg and clutch characteristics across all genera of living turtles is still missing. We present analyses based on data from the literature for at least one representative of each extant turtle genus, in order to identify trends in reproductive strategies and investigate potential factors that influence clutch and egg traits. We addressed the following questions: (1) Are reproductive traits (such as egg size, egg shape, and clutch size) correlated as predicted by egg size theory? (2) Are turtle species from different clades converging in their reproductive strategies? (3) Do ecological factors (such as distribution, and diet) influence egg/clutch characteristics? We hypothesize that reproductive traits in turtles evolved independently several times. Furthermore, we hypothesize that egg and clutch characteristics follow the predictions of egg size theory. Such characteristics are influenced by several ecological factors (*e.g.*, carnivores tend to produce bigger eggs and tropical species tend to produce bigger clutches).

## MATERIALS AND METHODS

We collected morphological (carapace size), ecological (climatic zone and diet) and reproductive data (egg size, clutch size, and number of clutches per year) for at least one species of each turtle genus (Table 1; Appendix S1) using available literature. We used Google scholar to perform an electronic search using different combinations of the key words "Egg size", "turtle reproduction", "breeding", "nest", "clutch size", "egg width". Studies from all dates were considered, as evolutionary characteristics of species do not usually change within the relevant time for a literature search. Only full-text reports in English, Spanish and Portuguese were considered. Study eligibility was assessed by one investigator. A secondary search was conducted on the reference list of these publications as well as on the list of publications that have cited the previous accessed one. The search continued until the limit of four articles containing information on the same ecological data for each species. The search was conducted following PRISMA (*Moher et al., 2011*) guidelines (Appendix S2).

**A**

Phylogenetic tree mapped with turtles clutch sizes

**B**

Phylomorphospace plot of turtle egg strategies

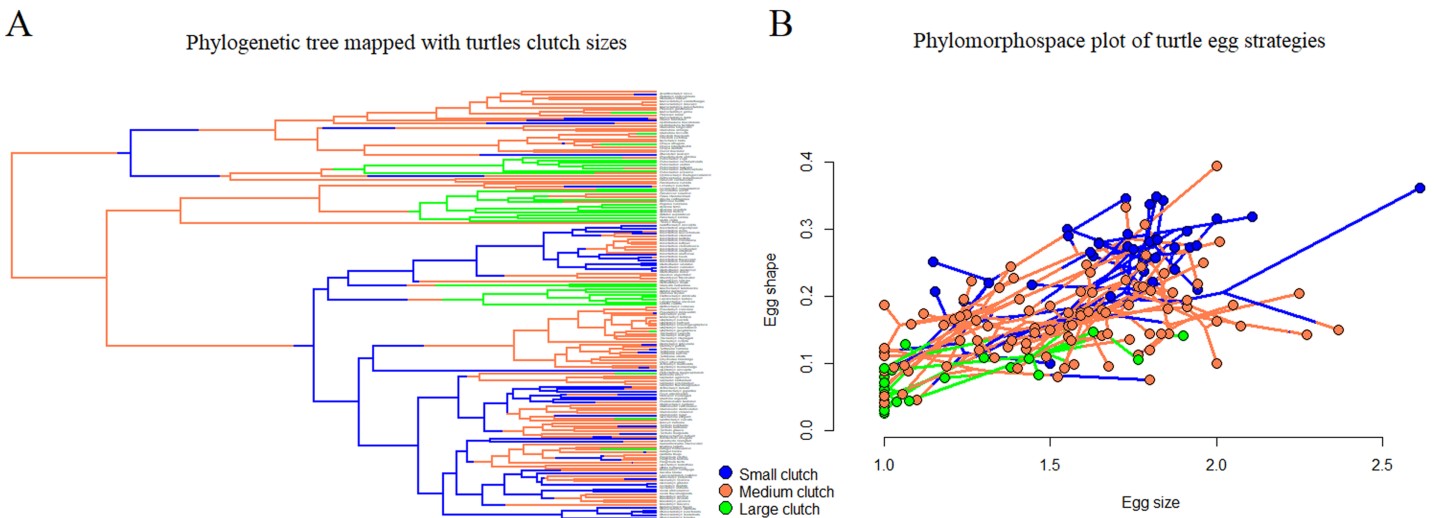

**Figure 2 Distribution of egg and clutch traits in the turtle phylogeny.** Different clutch sizes were assigned to three different regimes (small, medium, and large) and mapped to the tree (A); turtle phylogeny was plotted in a morphospace based on egg size and shape (B).

Data used was based on a combination of the information available (*e.g.*, smallest and biggest clutch sizes reported, even if from different sources) or based on the most common attribution for each species (e.g., species that are found both in land and water but most commonly in water were addressed with this kind of habitat). Data from captivity was considered as the characteristics of interest are mostly supposed to be inheritable and we considered possible bias to be irrelevant.

All statistical and exploratory analyses were conducted in the R statistical environment (v.4.0.4) (R Core Team; scripts and input files available in Appendix S3). We pruned the phylogeny proposed by *Pereira et al. (2017)* to match our dataset (Appendix S1) and used it in all the following analyses. Although this is not the most recent phylogeny available, it was the one with the biggest overlap with our dataset. Anyhow, we also ran the analyses using the most recent published phylogeny (*Thomson, Spinks & Shaffer, 2021*), and got the same results (Appendix S4).

### Are reproductive traits correlated as predicted by egg size theory?

In order to explore the correlation among reproductive parameters (egg width, egg length, and clutch size) commonly explored in previous works with smaller datasets (*Elgar & Heaphy, 1989*; *Iverson, 1992*; *Iverson et al., 1993*; *Rowe, 1994*), we fitted a hierarchical series of models to test for heterogeneity in the evolutionary rates and correlation of quantitative traits assigned to the tree (*Revell, Toyama & Mahler, 2021*). We used the function *evolvcv.lite()* from the R package Phytools (*Revell, 2012*).

We mapped the phylogeny through ancestral state reconstruction using traits based on clutch size (Fig. 2A), and the function *make.simmap()* from phytools R package: A. from 1 to 4 eggs, B. from 5 to 29 eggs, and C. 30 or more eggs. These groupings were arbitrarily chosen, and represent discrete traits of maximal clutch size among turtles. As continuously valued traits, we used egg length/carapace as a proxy for relative egg size

(ESI) and egg length/egg width as a proxy for egg "shape" (ESH). The choice for the model that better explains our analysis was based on Akaike information criterion (AIC; *Akaike, 1974*). We also plotted the data in a phylomorphospace, to support visualization (Fig. 2B).

### Do turtles have convergent reproductive strategies?

To test the hypothesis of convergence on reproductive traits among turtles, we calculated the angle between tree terminals assigned with similar multivariate phenotypic data (computed as the inverse cosine of the ratio product, and the product of vectors sizes), which represents the correlation coefficient between the terminals (vectors of the calculated angle), that represent a measure of phenotypic resemblance. we used the function *search.conv()*, from the R package RRphylo (*Castiglione et al., 2019*).

Small angles between vectors imply similar phenotypes, while angles around 90° and 180° represent dissimilar and opposing phenotypes, respectively (*Castiglione et al., 2019*). To verify convergence, we test if the differences between groups are smaller than expected considering their phylogenetic distance. The function can be used to test convergence either over entire clades or among species assigned to different states. Considering that our hypothesis of convergence includes species scattered along the phylogeny in a complex evolutionary history, a search without predetermined groups would have to compare the angles of phenotypic divergency between all species combinations (oppositely to comparisons among clades with higher taxonomic level such as families). Therefore, we assigned states based on clutch size (Appendix S1) to ensure computational viability. This decision also follows the methodology implemented in the previous question, and is based on the observations that turtles with larger clutches tend to produce smaller and rounder eggs, and possess large body size, while turtles that produce small clutches tend to show larger, elongated eggs, and have a small body size.

We first ran the analysis by testing if cryptodirans and pleurodirans converge in their reproductive strategies. To do that, we assigned each species to one of six different states: pleurodirans that produce clutches containing A. below five eggs, B. from five to 29 eggs, and C. 30 or more eggs; or cryptodirans that produce clutches containing C. below 5 eggs, D. from 5 to 29 eggs, and E. 30 or more eggs. We divided the same characters into two different states based on suborder (A and D, B and E, and C and F) in order to follow the analysis requirements. As it tests the convergence of groups distantly related, character states must be considered different. We used the suborders Cryptodira and Pleurodira to assign different characters as they are the most comprehensive taxonomic levels among turtles. By doing this, we tested if species in between these suborders are converging among three different states based on clutch size (small, medium and large). The convergence test between different traits in different clades represent the null models (they are not expected to converge).

Later, we ran a second analysis, without any separation among turtles, to test if species with small clutches (up to four eggs) and large clutches (over 30 eggs) diverge in their reproductive traits. The divergency test is nothing more than another convergence test, but opposing the reproductive traits hypothesized to diverge, which also works as our null

model (lack of convergence). Both tests were simulated 1,000 times and tips under the focal states were randomly removed until clustering remained with only three tips.

### Do ecological factors influence egg/clutch characteristics?

In order to estimate which ecological factors influence reproductive traits, we ran two different phylogenetic generalized least square (PGLS) models (*Grafen, 1989*; *Rohlf, 2001*; *Martins & Housworth, 2002*). In the first analysis, we tested how climatic zone, diet and the log mean clutch predict egg size.

In the second analysis, we tested how independent variables (climatic zone, diet and egg size) predicted the fecundity in turtles. We used the maximum number of eggs laid per clutch times the mean number of clutches per year as a proxy for fecundity.

Model diagnosis was performed for both tests (*Garamszegi, 2014*). We log-transformed the mean number of eggs per clutch (clutch mean) and fecundity to avoid skewed distribution of the predictor and to achieve homoscedasticity and normality of residuals (*Mundry, 2014*), respectively.

Multicollinearity between categorical predictors was tested using chi-squared tests (*Mundry, 2014*; *R Core Team, 2020*). We used maximum likelihood and Pagel's lambda model (*Pagel, 1997*, *1999*) to control for phylogenetic signal when fitting both PGLS. We used the function *gls()* of the package nmle (*Pinheiro et al., 2020*).

We used *P* values to infer which predictors significatively influence the model (*Symonds & Blomberg, 2014*; *Mundry, 2014*). We calculated each predictor's coefficient and its 95% confidence intervals using the PGLS scores table and the function *confint()*, respectively (*R Core Team, 2020*).

## RESULTS

The best fitting among all models used to test correlation among reproductive traits (highest log-likelihood scores, Table 1) was the "different rates, common correlation" model. Egg traits (ESH and ESI) coevolve and correlate in the same way with the regimes of traits mapped in the tree (number of eggs per clutch; $R = 0.567$), although with different evolutionary rates. Different regimes of clutch size occupy different regions of the morphospace (Fig. 2B).

The first convergence analysis revealed significative results in tests performed against same characters between different pleurodirans and cryptodirans (convergence test, $p = 0.001$, Table 2, in green). Additionally, the analysis also indicated significative results for convergence tests between medium sized clutches (B and E) and other size clutches (A, C, D, and F), although only between different suborders (convergence test, $p = 0.001$, Table 2, in bold). The divergency test failed to find any signs of convergence (convergence test, $p = 1.0$) between turtles with small and large clutches (Appendix S5).

In our test of the influence of ecological factor over egg/clutch characteristics, all the independent variables (climatic zone, diet and clutch mean) were significant in predicting egg size in turtle species in the first PGLS analysis (Table 3; Fig. 3). Egg size and climatic zone were significant predictors of fecundity in turtles (Table 4; Fig. 4).

**Table 2 Tests of convergence among different reproductive strategies in turtles.**

| State 1 | State 2 | *P* value |
|---|---|---|
| D | E | 1 |
| D | F | 1 |
| D | A | **0.001** |
| D | B | **0.001** |
| D | C | 0.482 |
| E | F | 0.703 |
| E | A | **0.001** |
| E | B | **0.001** |
| E | C | **0.001** |
| F | A | 0.229 |
| F | B | **0.001** |
| F | C | **0.001** |
| A | B | 0.195 |
| A | C | 0.754 |
| B | C | 0.133 |

**Note:**
Letters represent traits based in different clutch sizes: small (below five eggs), medium (from five to 29 eggs), and large (30 or more eggs), for pleurodirans (A, B, and C, respectively), and cryptodirans (D, E, and F, respectively). Tests that presented significative results for convergence (*P* = 0.001) are in bold. Tests between same traits and between same suborders are in green.

**Table 3 Phylogenetic generalized least squares scores of variables predicting egg size in turtles.**

| Predictor | Coefficient | SE | Lower CI | Upper CI | *p* value |
|---|---|---|---|---|---|
| Climatic zone | | | | | 0.002 |
| – Temperate | 0.296 | 0.019 | 0.259 | 0.333 | |
| – Tropical | 0.288 | 0.028 | 0.234 | 0.342 | |
| Diet | | | | | <0.001 |
| – Carnivore | 0.296 | 0.019 | 0.259 | 0.333 | |
| – Herbivore | 0.278 | 0.031 | 0.217 | 0.340 | |
| – Omnivore | 0.306 | 0.031 | 0.245 | 0.367 | |
| Clutch mean | −0.056 | 0.004 | −0.064 | −0.049 | <0.001 |

**Note:**
Climatic zone, diet and clutch mean predict the size of the egg in turtle species. SE, standard errors. CI, confidence intervals.

## DISCUSSION

The evolutionary history of turtles is marked by a complex pattern of character evolution regarding their reproductive strategies (*e.g.*, changes in egg size, egg shape and clutch size). Our analyses support the interpretation of repeated changes in these characters over evolutionary history. The hypothesis that large-bodied turtles tend to produce larger clutches with comparatively smaller and rounder eggs, while small-bodied species produce small clutches with larger and more elongated eggs seems to be supported by general patterns described in both the analyses here as well as those in previous literature (*Elgar &*
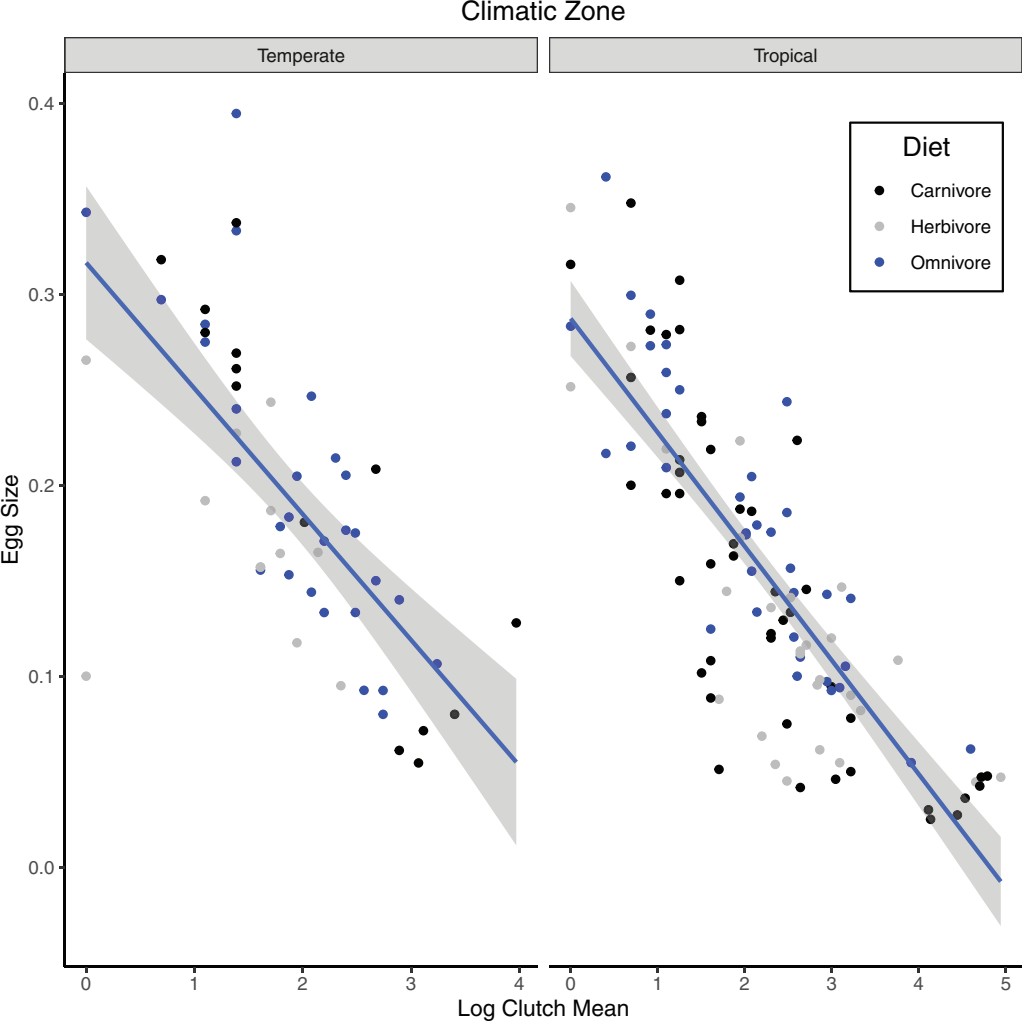

**Figure 3 Phylogenetic generalized least squares model of variables predicting egg size in turtles.** The model predicts the relationship of relative egg size (egg length/carapace length) to log mean clutch size (mean number of eggs laid per clutch) for turtle species that occupy different climatic zones (temperate or tropical) and have different diet types (carnivory, herbivory, or omnivory).

**Table 4 Phylogenetic generalized least squares scores of variables predicting fecundity in turtles.**

| Predictor | Coefficient | SE | Lower CI | Upper CI | *p* value |
|---|---|---|---|---|---|
| Climatic zone | | | | | 0.005 |
| – Temperate | 5.104 | 0.353 | 4.413 | 5.796 | |
| – Tropical | 5.247 | 0.503 | 4.263 | 6.232 | |
| Diet | | | | | 0.378 |
| – Carnivore | 5.104 | 0.353 | 4.413 | 5.796 | |
| – Herbivore | 4.970 | 0.569 | 3.856 | 6.085 | |
| – Omnivore | 5.255 | 0.564 | 4.150 | 6.360 | |
| Egg size | −11.426 | 0.907 | −13.203 | −9.649 | <0.001 |

**Note:**
Climatic zone and egg size predict fecundity in turtle species. SE, standard errors. CI, confidence intervals.

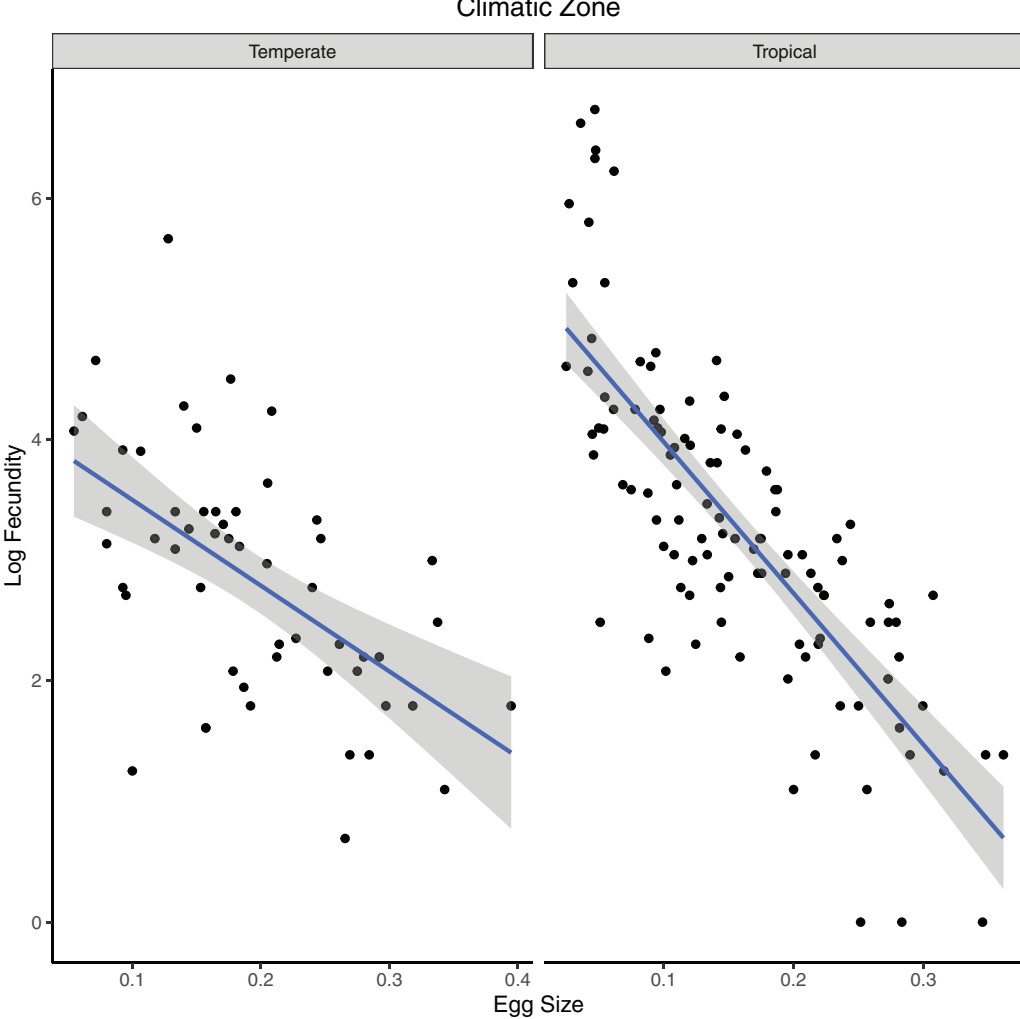

**Figure 4 Phylogenetic generalized least squares model of variables predicting fecundity in turtles.**
The model predicts the relationship of log fecundity (maximum number of eggs laid per clutch times
the mean number of eggs laid per clutch) to relative egg size (egg length/carapace length) for turtle species
that occupy different climatic zones (temperate or tropical).

*Heaphy, 1989*; *Iverson, 1992*; *Iverson et al., 1993*; *Iverson, Lindeman & Lovich, 2019*; *Rowe,
1994*; *Rachmansah, Norris & Gibbs, 2020*).

The results of our analysis are consistent with the predictions of the egg size theory.
The selected model reflects a tendency for the traits "egg size" and "egg shape" to positively
coevolve, while both are inversely correlated to "clutch size". Moreover, based on the
distribution of characters along the tree and in the phylomorphospace, these patterns
evolved independently and recurrently along the diversification of turtles. During their
evolutionary history, turtles explored different reproductive strategies with several
instances of convergent evolution.

However, correlation does not imply causation and the interpretation of observed
patterns as an example of evolutionary convergence is not straightforward (*Kluge, 2005*;

*Stayton, 2015*). To be able to make inferences about evolutionary patterns, we used quantitative measures and falsified our hypotheses with null models (*Popper, 1982*; *Stayton, 2015*) through the function *search.conv()* (*Castiglione et al., 2019*). Turtles of the Pleurodira and Cryptodira converge in all three different reproductive strategies tested.

Because all traits used in the convergence tests are continuous, without any clear break in the patterns, we also recovered significative results among medium size clutches and small or big size clutches between different "suborders". The fact that the same results were not recovered within suborders, is an indication that this result is a type I error. Because "suborder" is a highly inclusive taxonomic rank, the phenotypic differences between groups are considered smaller than expected considering their evolutionary distance. When the same traits are tested within the same "suborder", their differences are not less than expected and, therefore, are not pointed as convergent.

The divergence test between small and large clutches among all turtles indicated these traits are indeed divergent, independently of the taxonomic rank. Therefore, evidence indicates that turtles are converging to at least two—most probably three—different egg and clutch strategies, with continuous traits that prevent a clear differentiation among them. Nevertheless, these traits follow the egg/clutch size theory.

Although our analyses provided evidence for convergence among different turtle clades, they do not explain the reasons for such convergence. With the PGLS analyses, we were able to specify some of the ecological pressures affecting egg and clutch characteristics. Climatic zone is the only factor to partially explain the correlations in both analyses. Egg size is also influenced by diet with herbivores producing relatively smaller eggs. Tropical species have smaller eggs and a higher mean number of eggs per clutch compared to species from temperate areas (similar results have been reported for specific turtle clades in previous works, see *Macip-Ríos et al. (2017)* for an example in Kinosternidae). Additionally, high protein intake seems to stimulate egg production in turtles and other animals (*Watanabe et al., 1984*; *Bjorndal, 1985*). These results might be influenced by the broad availability of resources in tropical areas, enabling larger-bodied taxa that can produce more eggs (*Rachmansah, Norris & Gibbs, 2020*). It might also play a role in egg nutrition (*Craven et al., 2008*) and in favoring earlier hatching, as tropical areas have higher mean temperatures throughout the year, which increases metabolism in embryos (*Spencer & Janzen, 2011*).

Since many turtle families are spread across different geographic areas, in different climatic zones and with specific available resources (*e.g.*, Emydidae across the Americas; *Bour, 2007*), closely related species are subjected to extremely different ecological pressures. Sympatric distant related species, however, suffer similar ecological pressures and, therefore, tend to occupy similar ecological niches despite their intrinsic phylogenetic distance (*Kim, 2016*).

Aside from the importance of ecological factors in egg and clutch characteristics, the PGLS analyses also support our first analysis on egg and clutch correlations. There is a negative correlation between relative egg size and clutch size, demonstrating that reproductive traits are correlated as predicted by egg size theory. Based on these results, we conclude that there are major trends in reproductive strategies to which turtles converge.

In addition to the traits tested in the present study, other factors may play important roles in egg and clutch strategies of turtles, and could contribute to shaping the patterns found in our analyses. Adaptations within specific niches are worth mentioning and should not be forgotten when interpreting this complex scenario (see *Kluge, 2005* and *Losos, 2011* for a review of the role of convergent evolution in inferring adaptations). For instance, little is known about many of the aspects that influence the reproductive characteristics within Testudines. These include specific environmental pressures (as suggested by *Hofmeyr, Henen & Loehr (2005)* for *Homophus signatus*; and by *Hedrick et al. (2018)* for *Chelydra serpentina*, the last case within annual changes over the same population), patterns of reproductive allocation within and among species (*Wilkinson & Gibbons, 2005*), morphological constraints (*Lovich et al., 2012*), conflicts in parent-offspring size (*Janzen & Warner, 2009*), anti-predatory strategies (*Santos et al., 2016*), maternal effects and parental care (*Hughes & Brooks, 2006*; *Warner, Jorgensen & Janzen, 2010*).

As mentioned by *Nussbaum (1987)*, the safe harbor hypothesis suggests that parental care makes the embryonic stage the safest harbor, favoring egg size to increase in species with parental care, and consequently decreasing the duration of sequential stages with higher risk. Testudinidae is the turtle clade with the largest number of species known to care for their eggs (*Agha et al., 2013*). Although still an uncommon behavior within Testudinidae, it makes the safe harbor hypothesis a possible explanation for the comparatively larger eggs and, consequently, smaller clutches in most species of this clade.

Although other turtle clades have historically been considered to lack any form of parental care, there is now evidence to the contrary (*Ferrara, Vogt & Sousa-Lima, 2013*). The Arrau turtle (*Podocnemis expansa*) is the biggest South American freshwater turtle, and produces many small round eggs in a clutch. In this case, the only described parental care behavior starts after the eggs hatch, providing the safe harbor hypothesis with only weak explanatory power. Other factors probably have a bigger influence in this case, such as the proposition that round eggs suffer less from desiccation (*Elgar & Heaphy, 1989*; *Hofmeyr, Henen & Loehr, 2005*).

As noticed by *Elgar & Heaphy (1989)*, terrestrial species lay larger eggs in smaller clutches compared to freshwater or marine species, but this is a statistically confounded association because of the fact that turtle families represent ecological groups. The convergent distribution of reproductive traits and the different modifications of these traits across families that occupy unique niches—such as Testudinidae that live on land and Cheloniidae/Dermochelyidae that live in the ocean—could be considered evidence for the adaptation of specific clades to an "optimal" reproductive strategy in a specific environment or under a specific constraint.

The fact that the distribution of these strategies is associated with groups that colonized new environments provides strong support for a heuristic assumption of adaptive value (*Kluge, 2005*; *Losos, 2011*; *Thomson, Spinks & Shaffer, 2021*). At the same time, asserting the adaptive value of some of these traits can be difficult (see *Kluge, 2005*), and the correlation between specific traits and families that form ecological groups prevents the

postulation of statistically supported tests, which makes hypotheses based on niche adaptations greatly speculative (*Popper, 1982*; *Stayton, 2015*).

## CONCLUSIONS

We conclude that egg and clutch traits in Testudines evolved independently several times across non-directly related clades that converged to similar reproductive strategies.

Egg and clutch characteristics follow the trade-offs predicted by egg size theory and are influenced by ecological factors. Climatic zone plays an important role in the distribution of reproductive characteristics among turtles, and diet influences egg size.

## ACKNOWLEDGEMENTS

We thank Dr. Richard Vogt for providing important literature on egg/clutch diversity, Anne-Claire Fabre, Julien Clavel, Danilo Muniz and Diogo Melo, for helping with exploratory analyses, and Ana Balcarcel for revising the manuscript. We also thank the anonymous reviewers and the editor, Diogo Provete, for all the work done on improving this manuscript.

### Funding

Coordenação de Aperfeiçoamento de Pessoal de Nível Superior (CAPES) supported Pedro Henrique Dias (Proc. 88887.364687/2019e00). Swiss Government Excellence Scholarship (ESKAS) supported Gabriel Jorgewich-Cohen (Grant number: 2020.0190). This work was supported by SNF Grant No. 31003A-169395 to Marcelo R Sanchez-Villagra and by the Federal Commission for Scholarships for Foreign Students (FCS, Switzerland) to Gabriel Jorgewich-Cohen. The funders had no role in study design, data collection and analysis, decision to publish, or preparation of the manuscript.

### Grant Disclosures

The following grant information was disclosed by the authors:
Coordenação de Aperfeiçoamento de Pessoal de Nível Superior (CAPES): 88887.364687/2019e00.
Swiss Government Excellence Scholarship (ESKAS): 2020.0190.
SNF Grant: 31003A-169395.
Federal Commission for Scholarships for Foreign Students (FCS, Switzerland).

### Competing Interests

The authors declare that they have no competing interests.

### Author Contributions

- Gabriel Jorgewich-Cohen conceived and designed the experiments, performed the experiments, analyzed the data, prepared figures and/or tables, authored or reviewed drafts of the paper, and approved the final draft.

- Rafael S. Henrique conceived and designed the experiments, performed the experiments, analyzed the data, prepared figures and/or tables, authored or reviewed drafts of the paper, and approved the final draft.
- Pedro Henrique Dias analyzed the data, authored or reviewed drafts of the paper, and approved the final draft.
- Marcelo R. Sanchez-Villagra conceived and designed the experiments, analyzed the data, authored or reviewed drafts of the paper, and approved the final draft.

## Data Availability

The raw data from literature and treated data for model analysis and codes are available in the Supplemental Files.

## Supplemental Information

Supplemental information for this article can be found online at http://dx.doi.org/10.7717/peerj.13014#supplemental-information.

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
