# Peer review of "The evolution of reproductive strategies in turtles"

_PeerJ, doi:10.7717/peerj.13014_

## Round 0.1 · original submission · Major Revisions

I have now received one review about your work recently submitted to PeerJ. It's been extremly hard to find reviewers lately, which is exacerbated during the holidays. I have contacted 8 people to get one review. To speed up the editorial process and to give you more time to work on the revision, I decided to send the decision with only this reviewer comments and my own.
That having said, I thought your paper has great potential, due to the questions and data you used. However, most of the data analysis is not correct and should be re-done completely. There're also many theoretical misconceptions along the text that should be resolved. First, you frame your question in terms of "optimal egg strategy". This evolkes an Ornstein-Uhlenbeck (OU) model of trait evolution. Nonetheless, you don't use any method to test it nor talk about any of the theory behind it. I highly suggest you to read through the literature on phylogenetic comparative methods, specially those papers refering to the OU model (Felsenstein 1988, Butler & King 2004, Cooper et al. 2016, Ho & Ané 2014, Meara & Beaulieu 2014 chapter) and then frame your question in terms of this model, For example, does habitat exerts a selective pressue on mean clutch size? Several R packages are available to deal with OU: OUwie, bayou, geiger, mvMORPH etc. You have a beautiful data set at hand, but have to better use it. This mind map can be of some help https://coggle.it/diagram/WhbkkxE2BAAB0R0m/t/summary-of-phylogenetic-comparative-methods-diogo-b-provete

There're other problems with data analysis. First, you said you use a "model selection". However, after carefully chacking your R script and the M&M, I saw that what you did was actually some kind of stepwise selection (along with data dredging), which is an exploratory tool to select *variables*, not models. Also, even having a dated phylogeny at your disposal you used a linear mixed-effects model in the stepwise procedure, including family as a random factor. This is not correct, since you had to use an explicitely phylogenetic model, such as PGLS (see the attached pdf).

Given the problems with data analysis, the discussion will have to be fully re-written. Avoid citing figures and tables in the discussion.

All my detailed comments are in the attached pdf. I have also played a bit with your R script and dataset (thank you for having submitted those) and provided some guidance on how to analyse the data in a more correct way in a compile R markdown dynamic document in the same pdf, following the manuscript. The source .Rmd file can be found at

https://drive.google.com/file/d/1SjvEqF_qv1O707CMlgSOqZB4K6IlqJ63/view?usp=sharing

·

Basic reporting

First of all, there is a lack of literature up to date and most of the manuscript rely upon old bibliography. Despite important, such old literature (1980 decade) were overlapped by new theories. For instance, there are several bibliography on turtles geographical distribution X climate or phylogenetic associated to body size and/or behavior (for instance: de la Fuente, M. et al. 2014. Origin, Evolution and Biogeographic History of South American Turtles; Angielczyk, K.D., Burroughs, R.W. & Feldman, C.R. 2015. Do turtles follow the rules? Latitudinal gradients in species richness, body size, and geographic range area of the world's turtles. J. Exp. Zool. B Mol Dev Evol. 2015 May;324(3):270-94. doi: 10.1002/jez.b.22602; Rodrigues, J.F.M. & Diniz-Filho, J.A.F. 2016. Ecological opportunities, habitat, and past climatic fluctuations influenced the diversification of modern turtles. Mol. Phyl. Evol. 101: 352-358). Also, authors must search for literature reviews on the theme to have a more robust theoretical approach. Otherwise, it remains most speculative (for instance: Janzen, F.J. & Warner, D.A. 2009. Parent-offspring conflict and selection on egg size in turtles. J Evol Biol. 2009 Nov;22(11):2222-30. doi: 10.1111/j.1420-9101.2009.01838.x. Epub 2009 Sep 29. PMID: 19796084; Iverson, J. B., Lindeman, P. V., & Lovich, J. E. (2019). Understanding reproductive allometry in turtles: A slippery "slope". Ecology and evolution, 9(20), 11891–11903. https://doi.org/10.1002/ece3.5697).
Figure 2 must be improved. Points from Figure 4 must be related to solid lines. Figure 1 is not informative at all (please, delete).

Experimental design

It lacks a more detailed hypothesis explanation as well as its premises.
Even the supplementary material must be reviewed to introduce and/or change mistake information on turtles reproductive biology.

Validity of the findings

The discussion section must be improved by explaining what really your results tell us about your idea. For instance, what means those phylogenetic trees for each variable throughout evolutionary history?

Additional comments

The manuscript "The evolution of reproductive strategies in turtles" address simple but interesting points and aim to verify which factors shape an important issue on turtles evolutionary history. Despite such approach has been investigated in the last years, the authors fail to go deep in the theme.

---

## Round 0.2 · Major Revisions

I have received two reviews of your work and they were pretty much complementary to each other. While this version of the paper is certainly improved in relation to the previous one, the reviewers provided a number of critiques that must be addressed, which is why you're receiving another 'major revision' decision. I also spotted a some problems in the Methods. R1 mentioned that some key papers about egg size and shape evolution are missing and recommended that you incorporate them in the paper. R2 points out that you didn't actually test for 'convergence' in your paper. (s)he recommends some methodological papers, but there're newer ones, such as

https://besjournals.onlinelibrary.wiley.com/doi/full/10.1111/2041-210X.12195

https://onlinelibrary.wiley.com/doi/full/10.1111/brv.12257

SURFACE is quite problematic (see Uyeda & Harmon 2014 and others) so, I don't recommend using it. If you do want to dive into clade-specific methods for testing convergence in univariate traits, check the packages bayou or l1ou and papers by Natalie Cooper and Cécile Ané on OU model.

Try to improve Fig. 2, it's not as appealing as it could be. Color legend at the bottom is pretty small, as well as the names on it.

Reviewer 2 ·

Basic reporting

I complete the review of the manuscript (MS) “The evolution of reproductive strategies in turtles” submitted by Jorgewich-Cohen et al. The MS try to test the hypothesis of convergence of reproductive traits using a wide data base covering all turtle genera. Authors approached the data analysis using comparative methods and at least two evolutionary models (OU and Brownian motion). The data set is interesting, and they used relevant variables, they compared between elongated eggs vs. spherical eggs, climate zones, and diet as categorical variables, using clutch size, fecundity, and egg size as continuous variables. I also reviewed the response letter to the comments made on their first round of revision. In my opinion, this version improves a lot of sections of the MS and focus the analysis on the evolution of reproductive strategies in an evolutionary convergence context. Nevertheless, I consider the MS still confusing and I´m not sure if they really test evolutionary convergence in reproductive strategies. I fund several inconsistencies along the MS and I consider authors avoid important references on the evolution of life history of turtles.

Experimental design

The idea of the paper is very interesting; however, their analysis and interpretation is confusing. I think authors are not familiarized with the methods to test convergence. I suggest pertinent literature directly in the MS. One of the important flaws of this paper is there is a lack of seminal papers in their literature review. Authors omit other studies trying to explain the life-history evolution in turtles such those of Wilbur and Morin (1988), and Smith and Fretwell (1974) on optimal egg size theory, among others. Otherwise, authors omit important approaches on the study of convergence such those of McGhee (2011), among others.

Validity of the findings

I fund the following issues concerning and I think authors should resolve these comments before the MS has been consider for a new review:

Authors claim that the independent evolution of a trait or a character indicate convergence, this is not necessarily true, it could be parallelism. Authors should review McGhee, G.R. 2011. Convergent Evolution: Limited forms most beautiful. MIT Press, for a detailed and modern explanation of convergence and how approach it.

Authors motion the used Pereira’s et al. phylogeny, but they did not mention how the resolve the missing species in the phylogeny or their data set? Did they prune the phylogeny?

The results section is very brief. Authors make a lot of tests and used complex analytical tools; I think authors should expand their results report. At least not only refer the reader to the tables and figures. Figure 2 is ok, but it needs a color code inside the figure because it´s very hard to follow.

Authors mention the optimal egg size theory as theoretical guide of their work. Their entire framework is based on this theory, however, they didn´t mention the possibility of pelvic constraint as an alternative hypothesis. Both hypotheses have been proved in the past and there is evidence supporting both. Authors should acknowledge the evidence of pelvic constraint hypothesis, at least in small body size species or individuals within a single population.

I´m not sure authors provide proofs of convergent evolution. Perhaps they have a clue, or their data suggest a potential convergence, but they didn´t proof these turtles’ lineages were affected by similar environmental or more precisely, similar selection pressures to derive in convergent life history traits. Other thing, authors transform continuous variables to interval variables, which they map on the phylogeny. Since reproductive traits such clutch size, egg size, and fecundity vary yearly, seasonally, and even between females of the same population. A more important approach should include a variation measure such standard error or coefficient variation. Even if their data suggest or point to a possible convergence, authors should conduct other studies to really document this phenomenon.

Discussion has some flaws. Authors quoted other studies several times to contrast their results and to support his data, I think they should contrast their data with the framework they develop. Discussion didn’t support their claim of convergent evolution, also they mention several other factors could be driven the evolution of life history in turtles. Authors cited Johnathan Losos work on Anolis convergence, but authors didn´t follow the same methods and framework suggested by Losos (2011) and in other work conducted by Professor Losos.

Finally, I consider this paper still interesting, however the MS is not fully developed and still needs work and authors should integrate a more compete framework.

Additional comments

I complete the review of the manuscript (MS) “The evolution of reproductive strategies in turtles” submitted by Jorgewich-Cohen et al. The MS try to test the hypothesis of convergence of reproductive traits using a wide data base covering all turtle genera. Authors approached the data analysis using comparative methods and at least two evolutionary models (OU and Brownian motion). The data set is interesting, and they used relevant variables, they compared between elongated eggs vs. spherical eggs, climate zones, and diet as categorical variables, using clutch size, fecundity, and egg size as continuous variables. I also reviewed the response letter to the comments made on their first round of revision. In my opinion, this version improves a lot of sections of the MS and focus the analysis on the evolution of reproductive strategies in an evolutionary convergence context. Nevertheless, I consider the MS still confusing and I´m not sure if they really test evolutionary convergence in reproductive strategies. I fund several inconsistencies along the MS and I consider authors avoid important references on the evolution of life history of turtles.

The idea of the paper is very interesting; however, their analysis and interpretation is confusing. I think authors are not familiarized with the methods to test convergence. I suggest pertinent literature directly in the MS. One of the important flaws of this paper is there is a lack of seminal papers in their literature review. Authors omit other studies trying to explain the life-history evolution in turtles such those of Wilbur and Morin (1988), and Smith and Fretwell (1974) on optimal egg size theory, among others. Otherwise, authors omit important approaches on the study of convergence such those of McGhee (2011), among others.

I fund the following issues concerning and I think authors should resolve these comments before the MS has been consider for a new review:

Authors claim that the independent evolution of a trait or a character indicate convergence, this is not necessarily true, it could be parallelism. Authors should review McGhee, G.R. 2011. Convergent Evolution: Limited forms most beautiful. MIT Press, for a detailed and modern explanation of convergence and how approach it.

Authors motion the used Pereira’s et al. phylogeny, but they did not mention how the resolve the missing species in the phylogeny or their data set? Did they prune the phylogeny?

The results section is very brief. Authors make a lot of tests and used complex analytical tools; I think authors should expand their results report. At least not only refer the reader to the tables and figures. Figure 2 is ok, but it needs a color code inside the figure because it´s very hard to follow.

Authors mention the optimal egg size theory as theoretical guide of their work. Their entire framework is based on this theory, however, they didn´t mention the possibility of pelvic constraint as an alternative hypothesis. Both hypotheses have been proved in the past and there is evidence supporting both. Authors should acknowledge the evidence of pelvic constraint hypothesis, at least in small body size species or individuals within a single population.

I´m not sure authors provide proofs of convergent evolution. Perhaps they have a clue, or their data suggest a potential convergence, but they didn´t proof these turtles’ lineages were affected by similar environmental or more precisely, similar selection pressures to derive in convergent life history traits. Other thing, authors transform continuous variables to interval variables, which they map on the phylogeny. Since reproductive traits such clutch size, egg size, and fecundity vary yearly, seasonally, and even between females of the same population. A more important approach should include a variation measure such standard error or coefficient variation. Even if their data suggest or point to a possible convergence, authors should conduct other studies to really document this phenomenon.

Discussion has some flaws. Authors quoted other studies several times to contrast their results and to support his data, I think they should contrast their data with the framework they develop. Discussion didn’t support their claim of convergent evolution, also they mention several other factors could be driven the evolution of life history in turtles. Authors cited Johnathan Losos work on Anolis convergence, but authors didn´t follow the same methods and framework suggested by Losos (2011) and in other work conducted by Professor Losos.

Finally, I consider this paper still interesting, however the MS is not fully developed and still needs work and authors should integrate a more compete framework.

I made several comments along the MS and made direct suggestion to the authors.

Literature cited

Smith, C.C. and S.D. Fretwell. 1974. The Optimal Balance between Size and Number of Offspring. The American Naturalist 108:499–506.

Wilbur H.M. and P.J. Morin. 1988. Life history evolution in turtle. P. 387-439. In: Gans, C. and Huey, R.B. (Eds.). Biology of the Reptilia Volume 16 Ecology B. Defense and Life History. Alan R. Liss, New York.

McGhee, G.R. 2011. Convergent Evolution: Limited forms most beautiful. MIT Press. Boston.

Losos, J. 2011. Lizards in an Evolutionary Tree: Ecology and Adaptive Radiation of Anoles. University of California Press, Berkeley.

Annotated reviews are not available for download in order to protect the identity of reviewers who chose to remain anonymous.

Reviewer 3 ·

Basic reporting

This study approaches an interesting and important evolutionary question about turtles’ “optimal egg strategy”, whereas egg and clutch size evolves by an energetic trade-off, influencing their egg shapes. The authors studied the “optimal egg strategy” by applying established and modern phylogenetic comparative methods on a dataset comprehending all turtle genus. The paper has professional English, structure, figures, and tables. The raw data and analytical scripts were shared. I do not have the expertise to judge the literature references about "optimal egg strategy" in turtles, but it seems well represented. The introduction needs to explore and provide more information about how and which environmental and ecological variables (e.g. climatic zone, diet, habitat, and zoogeographical regions) determine egg and clutch sizes, as these variables are evaluated further in the methods. The reference misses articles about convergent evolution such as Ingram & Mahler (2013), Bastide et al. (2018), Castiglione et al. (2019), and Mahler et al. (2017). The paper is self-contained, but the statistical methods need to be better linked to which hypothesis they are evaluating. The manuscript has the potential to be published in PeerJ, but the authors need to improve some writing and methodological issues to have more robust statistical analyses and fluid writing, mainly in the methods and results.

Experimental design

This manuscript has ethical, well-defined, and relevant questions, within the aims and scope of the journal. The materials and methods section needs to be more straightforward in linking the statistical methods to the tested hypotheses. Also, the methods section requires more details about how data were collected, why the author chose the phylogenies they used, there are other published phylogenies comprehending 300 species (Rodrigues & Diniz-Filho, 2016) that may be useful. The fitting of multivariate evolutionary models requires details about the different regimes evaluated. For instance, there are regimes defined in Table 1 as control, diet, Clima, Zoogeography, but there is no explanation about these variables, how they were collected, how ancestral states were estimated, and which evolutionary models were used. Furthermore, some regimes were defined based on body size reconstruction, but the authors did not present it was done splitting body size into 2, 3, and 4 categories. These multivariate evolutionary models may describe how some variables evolved correlated to other variables (regimes), but it is not clear how it answers the question about convergent evolution (Mahler et al., 2017). Perhaps the use of methods developed for testing convergent evolution such as Bastide et al. (2018), Ingram & Mahler (2013), and Castiglione et al. (2019) may be better alternatives. MANOVA analysis also requires better explanations about what it is evaluating. Looking at the script, in supplementary material, it seems to test the effects of diet on reproductive traits, but the methods section says nothing about it.
The PGLS models included as independent variables climatic zone, diet, clutch size, and egg size, but in the introduction, body size is described as a well-known variable in determining both egg size and clutch size. Why body size was not included in the PGLS to control its effects on both response variables? Additionally, PGLS was evaluated by a Full-Null Model Comparison approach. It is not explained how this method works, there are no citations enabling the reader to understand the methodology. Furthermore, it is not clear why the authors excluded two species from the PGLS analyses, these species have wrong data, or the models used were not adequate to describe trait distribution? How these species' exclusion impacted ancestral state reconstruction and evolutionary model inferences? How the exclusion of these species impacted the analyses and conclusions of the manuscript?
A possible approach to improve this manuscript's understanding is fitting multivariate models to body size, egg size, egg shape, and clutch size to evaluate their correlations, thus answering the first paper’s question. Testing reproductive traits convergence with Bastide et al. (2018), Ingram & Mahler (2013), or Castiglione et al. (2019) methods. To answer the main question of the paper, optimal egg strategy, use PGLS models with lambda parameter to control phylogenetic signal (Revell, 2010).

References

Bastide, P., Ané, C., Robin, S., & Mariadassou, M. (2018). Inference of adaptive shifts for multivariate correlated traits. Systematic Biology, 67(4), 662–680. https://doi.org/10.1093/sysbio/syy005.

Castiglione, S., Serio, C., Tamagnini, D., Melchionna, M., Mondanaro, A., Di Febbraro, M., Profico, A., Piras, P., Barattolo, F., & Raia, P. (2019). A new, fast method to search for morphological convergence with shape data. PLoS ONE, 14(12), 1–20. https://doi.org/10.1371/journal.pone.0226949.

Ingram, T., & Mahler, D. L. (2013). SURFACE: Detecting convergent evolution from comparative data by fitting Ornstein-Uhlenbeck models with stepwise Akaike Information Criterion. Methods in Ecology and Evolution, 4(5), 416–425. https://doi.org/10.1111/2041-210X.12034.

Mahler, D. L., Weber, M. G., Wagner, C. E., & Ingram, T. (2017). Pattern and process in the comparative study of convergent evolution. American Naturalist, 190, S13–S28. https://doi.org/10.1086/692648.

Revell, L. J. (2010). Phylogenetic signal and linear regression on species data. Methods in Ecology and Evolution, 1, 319–329. https://doi.org/10.1111/j.2041-210X.2010.00044.x

Rodrigues, J. F. M., & Diniz-Filho, J. A. F. (2016). Ecological opportunities, habitat, and past climatic fluctuations influenced the diversification of modern turtles. Molecular Phylogenetics and Evolution, 101, 352–358. https://doi.org/10.1016/j.ympev.2016.05.025.

Validity of the findings

All data and scripts were shared to guarantee reproducibility. Statistical methods to evaluate convergent evolution need to be clear and more robust. The methodology implemented to evaluate Phylogenetic Linear Generalized Models (PGLS) needs more details to understand how variables were selected and how confidence intervals and coefficient of determination were calculated. Also, PGLS needs to control the effect of body size to estimate the partial effects of egg size and clutch size on each other.
The results of explanatory analyses are not clear, there is methodological information that appeared, for the first time, in the results. For example, Table 1 has information (e.g. control, zoogeography, body size, habitat) that did not appear before in the manuscript. It is not clear how MANOVA and the multivariate evolutionary models are complementary analyses. It is written MANOVA results are different from multivariate evolutionary models results, but they have different proposals, MANOVA tests the effect of diet on reproductive traits considering only tip data, multivariate models fit the models considering the ancestral reconstruction of other variables.
Conclusions are well stated, but as methodology and results are not robust and clear, it is not well supported by the results.

Additional comments

Some minor comments:

Line 34-43 Information about ecological and environmental variables may be included along lines 34 and 43.

Line 41 In this line it is written “These hypotheses, largely based on studies of trait correlations (Gibbons, 1982), can be tested by methods that consider phylogeny, as they reduce the variance of estimated regressions (Rohle, 2006).” This sentence seems out of context here.

Line 59-60 The second question is not clear enough. Rephrase this question to better represent what will be tested.

Line 62-63 The hypothesis is too vague. Define which ecological factors may impact egg and clutch size.

Line 67-72 Explain how PRISMA guidelines were conducted. Likely, it is better to write Appendix 2 here. However, even Appendix 2 requires more details about inclusion criteria.

Lines 74-92 Perhaps the exploratory analyses may be sent to supplementary materials in order to enhance readability.

Line 87-89 Create a paragraph describing the data collection and cleaning, synonym standardization, transformation of variables (ESI, ESH, FEC). Make clear that these variables are those modeled by the evolutionary models.

Line 108-113 Describe which variables had their ancestral state reconstructed to define evolutionary regimes, how reconstructions were done. Explain how body size was categorized to define its regimes.

Line 112 Instead "model fitness", use model selection.

Line 114-116 MANOVA is not testing regimes here, but the effect of diet on reproductive traits.

Line 131 Habitat and zoogeographic zones did not appear before in the manuscript.

Line 132-133 How variables collinearity was evaluated?

Line 135-138 Full-Null Model Comparison approach needs to be explained in detail and referenced. It is not clear how the coefficient of determination, statistical significance, and confidence interval were calculated.

Line 154 Diet means here an OUM model fitted over an ancestral state reconstruction of diet, as it was not described before in the methods section, it is causing confusion.

Line 182 Exclude the extra parenthesis.

Figure 1 change adult? to adult in the legend.

Figure 3 Make clear what are the types of climatic zones in the legend.

Table 1 Describe what column System means in the legend. Make clear that diet OUM is a diet regime and explain the other regimes.

---

## Round 0.3 · Major Revisions

I have now received back the comments of the same two reviewers of the previous round. Both were unanimous in mentioning that the methods section still needs to present more details of the analysis and literature search. I must say that I second their comments. The correct application of the PRISMA protocol requires you to provide as supplemental material a flow diagram saying how many papers you found and which ones were excluded, see http://www.prisma-statement.org/PRISMAStatement/FlowDiagram. So, please, include it in your Appendix 2

Also, R1 mentioned that while this paper was in review, a new and more comprehensive turtle phylogeny came out (https://www.pnas.org/content/118/7/e2012215118.short) and suggests you re-run the analysis with it to see if results change.

R2 also comments that it's not adequate to simply exclude two data points to improve residual distribution. The correct way to deal with it is to conduct sensitivity analysis. This could be easily done with the R package sensiPhy. I highly recommend you do it.
Needless to say that I also agree with R2 that mixing model selection with NHST is not adequate because they are based on distinct statistical strategies/philosophies. Since you already have presented the P values and interpreted them, stick with it only.

For the evol.vcv analysis, of which states are we talking about? For this analysis you have to define a priori the clades you expect to have a shift in the trait correlation. However, it seems you used categories of clutch size to do it. This doesn't make any sense because clutch size is one of your variabels of interest. You had to, e.g., use an independent variable here, such as habitat or diet to define the clades. Also which traits you actually used? The trade-off hypothesis was about egg size and number only, not adult body size.
The PCA plots of Fig 2 look pretty weird, with data points smashed, basically only PC1 matters. Is that correct?

Finally, PeerJ uses a structured abstract and I highly recommend you to follow it when submitting a revised version. The abstract itself needs to be revised since there are too few results described and most of this section is for methods and hypothesis.

I'm still interested in seeing your paper published, but you need to do a much better job in following the reviewer's and my recommendations.

Reviewer 2 ·

Basic reporting

The MS to test the hypothesis of convergence of reproductive traits using a comparative approach with a wide data base covering all turtle genera. This second version has been shortened and methods were reformulated. This version is improved and succinct, author did find the same conclusions overall.

Experimental design

Analitical approach is correct and did achieve the objectives and test the hypothesis of the study, how ever I consider the following points should be covered before acceptance:

1. I understand authors choose Pereira et al. (2017) phylogeny, but what about use the recent published phylogeny of Thomson et al (2021)? You could use both phylogenies and compare both results and see if you find any concordance between both phylogenies.
2. The average clutch size also could be used as a continuous variable, it could be transformed to log. I'm not sure if the categories of clutch size are valid. I understand they are arbitrarily chosen, but doesn´t make any sense the category A, why use a larger interval than A? Authors should try to analyze or re-think what you understand or consider of a "reduced clutch" or "large clutch".
3. Authors didn’t analyze the medium clutch size; they focus in large (>30 eggs) vs. small clutch size (<4 eggs), but what about the “middle” cutch mass? I think author should dig deep in the results on strategies on the middle spectrum of reproductive strategies.
4. I would like to see a clearer explanation or interpretation on the selection pressures inducing convergence in the reproductive strategies in turtles.

Validity of the findings

I think this paper is very interesting and find this version suitable for publication in PeerJ. However, I have some small concerns that authors should address before acceptance. I consider this concerns as minor changes.

In this new version of the manuscript authors really explain how they test the convergence in reproductive strategies in all genera of turtles.

Additional comments

I provide additional comments directly on the manuscript.

Annotated reviews are not available for download in order to protect the identity of reviewers who chose to remain anonymous.

Reviewer 3 ·

Basic reporting

This study (“The evolution of reproductive strategies in turtles”) approaches an interesting and important evolutionary question about turtles’ “optimal egg strategy”, whereas egg and clutch size evolves by an energetic trade-off, influencing their egg shapes. The authors studied the “optimal egg strategy” by applying established and modern phylogenetic comparative methods on a dataset comprehending all turtle genus. The paper has professional English, structure, figures, and tables. The authors shared the raw data and analytical scripts. I do not have the expertise to judge the literature references about "optimal egg strategy" in turtles, but it seems well represented. This paper is not self-contained as many methodological explanations are not described in the manuscript and need further searches in the literature.

The introduction is well-written and contextualizes the theory, but there is still lacking a better theoretical introduction about how the climatic zone and diet may affect “optimal egg strategy”. The authors cite that these variables may affect egg and clutch size, but they do not explain the process. It is missing in the methodological section important information about how methods were performed, such as convergence testing does not present the null model and how many times it was replicated. Basic information to guarantee reproducibility, even if the code to run the analyses is provided, should be included in the main manuscript.

Experimental design

This manuscript follows a high technical and ethical standard and is within the aims and scope of the journal. Like the previous version of the manuscript, there is yet a lack of methodological information. The implemented methodology can answer the questions that the authors intended to evaluate, but the description of statistical analyses and the presentation of the results are still confusing, lacking important information. For instance, the authors evaluated the correlation among some variables and tested if there are shifts in these correlations across the phylogeny, but they did not show the correlation, that is the main result to evaluate if clutch size and egg shape followed “the optimal egg strategy” expectation. Furthermore, the analyses tested correlation shifts based on categories of fecundity, it is not clear how it is related to the theory being evaluated. Also, the section describing the search of the literature is still vague, the authors did not present how many papers were found, where papers were searched, how papers were selected. It is not enough to say the literature search followed PRISMA guidelines, it must be described how it was done. The supplementary material shows more details about the literature search, and the inclusion of that description in the main manuscript would improve the paper.
The description of the methodology to evaluate convergent evolution is superficial, the readers may not understand how the methodology works. Also, they tested convergence between cryptodirans and pleurodirans, but it is also not clear how it is related to the “optimal egg strategy”. In other words, the methodology section needs to guide better the reader into how the methodology answers the questions and how they are related to the “optimal egg strategy”.

In PGLS test, two species were excluded from the analyses to achieve homoscedastic and normality of the residuals. The authors can perform a sensibility test to evaluate how the exclusion of the two species may affect the results and conclusions. It is a fast analysis and requires only re-run the function with the complete data. These two species are not wrong data, and may show a piece of important information about egg size and fecundity in turtles. For instance, these lineages may have a different evolutionary process than in the whole clade.

The authors mixed model selection and test of significance to evaluate the PGLS models, maybe it is better to exclude model selection as the results and variable’s importance were evaluated by significance tests. The model selection seems meaningless in these regressions.

Validity of the findings

The conclusion needs to be better sustained by the results. It is not clear enough how the results corroborate the “optimal egg size strategy”. The result shows that clutch size is inversely related to egg size, but there is no information about egg shape. Also, the authors evaluated and found convergence on clutch size evolution, but it cannot be extrapolated to “reproductive traits in Testudines evolved independently several times across non-directly related clades”. The answer to “optimal egg strategy” is on the correlation test that should be better explored. The authors only show that the correlations and rates of reproductive traits are constant across the phylogeny, the correlations among these variables should be presented. Perhaps, the shifts included in the correlation test may follow the shifts of the regimes found in the convergent evolution analysis, instead of the arbitrary categories that were used. Firstly, detect the shifts in clutch size evolution, then evaluate if there are correlations among the variables following the “optimal egg strategy”.

Additional comments

Line 41 – explain how environmental factors (climate zone), diet and biogeography affect clutch and egg size.

Line 89-90 - what is Model hierarchy selection, cite the paper describing the method and also cite AIC.

Line 124 – It is not Pagel correlation, it is known as Pagel’s lambda model. Make clear that is was done to control phylogenetic signal among species.

Line 188-189 – How niche should be taken into account, explain better it.

Line 191-197 – This paragraph needs a better link to the other paragraphs.

Line 219 – Where is the fact that reproductive strategies are correlated to colonization of new environments?

Line 221-225 – It is not clear the point in this sentence.

---

## Round 0.4 · Minor Revisions

One of the previous reviewer agreed to comment again on the manuscript. The other one just replied to my invitation saying (s)he has no more comments to make.

This version of the manuscript is by far the best we have so far. So, thank you for your effort in improving it following comments.
However, I have to agree with R3 that some aspects of the Methods still need some work. I have made a few comments in the pdf attached and I'd also ask you to follow R3 comments, which I hope will be our final round of review. The discussion also needs some attention.

In response to your comments in the rebuttal letter, I have to say that:

1) Yes, include the analysis with Thompson's tree in the supplemental materials

2) The abstract can still be improved. Notice that it doesn't adhere to the structured abstract PeerJ uses, with sections highlighted. Besides, it has many more introductory sentences than it needs. Restrict introductory sentences to two or three at most (it has 6 now) and make the largest part of the abstract results and discussion, with some methods outlined

Reviewer 3 ·

Basic reporting

This study (“The evolution of reproductive strategies in turtles”) approaches an interesting and important evolutionary question about turtles’ “optimal egg strategy”, whereas egg and clutch size evolves by an energetic trade-off, influencing their egg shapes. The authors studied the “optimal egg strategy” by applying established and modern phylogenetic comparative methods on a dataset comprehending all turtle genus. The paper has professional English, structure, figures, and tables, but requires more attention on typos. The authors shared the raw data and analysis scripts. I do not have the expertise to judge the literature references about "optimal egg strategy" in turtles, but it seems well represented.

The introduction is well written and contextualizes the theory. The lack of information in the methods section has been solved, but there are still unclear sentences.

Experimental design

This manuscript follows a high technical and ethical standard and is within the aims and scope of the journal. The authors improved the previous versions of the manuscript solving the lack of methodological information. The implemented methodology can answer the questions that the authors intended to evaluate, but the description of the convergence and trait correlation analyses is still confusing. For instance, the author wrote they performed a likelihood ratio test and Akaike information criterion to select the best model of traits correlation, but only presented the AIC table. Also, it is unusual selecting a model based on two different inferential philosophies (frequentist and model selection).

Validity of the findings

The conclusion needs to be better sustained by the results. Convergence analyses are still unclear and hard to interpret and achieve a clear conclusion.

Additional comments

133 – states instead of estates.

line 153- The null model should describe the null model used to test the statistical significance of the convergence test instead of the theoretical expectation.

Lines 175-178 – The description of variable’s selection is confusing.

Line 211-214 – This paragraph is too vague and hard to understand its importance to the discussion.

---

## Round 0.5 · accepted · Accept

Thank you for providing answers to these final comments. I’m glad to recommend this version of the manuscript for publication. Congratulations